# Dedifferentiated Endometrial Carcinoma Could be A Target for Immune Checkpoint Inhibitors (Anti PD-1/PD-L1 Antibodies)

**DOI:** 10.3390/ijms20153744

**Published:** 2019-07-31

**Authors:** Ruriko Ono, Kentaro Nakayama, Kohei Nakamura, Hitomi Yamashita, Tomoka Ishibashi, Masako Ishikawa, Toshiko Minamoto, Sultana Razia, Noriyoshi Ishikawa, Yoshiro Otsuki, Satoru Nakayama, Hideyuki Onuma, Hiroko Kurioka, Satoru Kyo

**Affiliations:** 1Department of Obstetrics and Gynecology, Shimane University School of Medicine, 6938501 Izumo, Japan; 2Department of Organ Pathology, Shimane University School of Medicine, 6938501 Izumo, Japan; 3Department of Pathology, Seirei Hamamatsu General Hospital, 4308558 Hamamatsu, Japan; 4Department of Obstetrics and Gynecology, Seirei Hamamatsu General Hospital, 4308558 Hamamatsu, Japan; 5Department of Pathology, Shimane Prefectural Central Hospital, 6938555 Izumo, Japan; 6Department of Obstetrics and Gynecology, Shimane Prefectural Central Hospital, 6938555 Izumo, Japan

**Keywords:** dedifferentiated endometrial carcinoma, mismatch repair deficient, microsatellite instability, endometrial cancer, immune checkpoint inhibitor

## Abstract

Dedifferentiated endometrial carcinoma (DDEC) is defined as an undifferentiated carcinoma admixed with differentiated endometrioid carcinoma (Grade 1 or 2). It has poor prognosis compared with Grade 3 endometrioid adenocarcinoma and is often associated with the loss of mismatch repair (MMR) proteins, which is seen in microsatellite instability (MSI)-type endometrial cancer. Recent studies have shown that the effectiveness of immune checkpoint inhibitor therapy is related to MMR deficiency; therefore, we analyzed the immunophenotype (MMR deficient and expression of PD-L1) of 17 DDEC cases. In the undifferentiated component, nine cases (53%) were deficient in MMR proteins and nine cases (53%) expressed PD-L1. PD-L1 expression was significantly associated with MMR deficiency (*p* = 0.026). In addition, the presence of tumor-infiltrating lymphocytes (CD8+) was significantly associated with MMR deficiency (*p* = 0.026). In contrast, none of the cases showed PD-L1 expression in the well-differentiated component. Our results show that DDEC could be a target for immune checkpoint inhibitors (anti PD-L1/PD-1 antibodies), especially in the undifferentiated component. As a treatment strategy for DDEC, conventional paclitaxel plus carboplatin and cisplatin plus doxorubicin therapies are effective for those with the well-differentiated component. However, by using immune checkpoint inhibitors in combination with other conventional treatments, it may be possible to control the undifferentiated component and improve prognosis.

## 1. Introduction

Dedifferentiated endometrial carcinoma (DDEC) is a rare and more aggressive type of endometrioid carcinoma than high grade endometrioid carcinoma [1,2,3]. In 2006, Silva et al reported cases of endometrial carcinoma in which low-grade endometrioid carcinoma was combined with undifferentiated carcinoma and designated them as dedifferentiated endometrial carcinoma [3,4]. A total of 50%–58% of patients with DDEC present with advanced stage disease, and 40% of these patients die within half a month to 20 months from the disease [3,5]. For this reason, it is urgently required to develop therapies, such as immunotherapy, that fit molecular subgroups.

DDEC was suggested to be related to the deficiency of mismatch repair (MMR) proteins, mutL protein homolog 1 (MLH1), postmeiotic segregation increased 2 (PMS2), mutS protein homolog 2 (MSH2), and mutS protein homolog 6 (MSH6), resulting in microsatellite instability (MSI) [5]. MMR deficiency has been reported in 58% of cases by immunohistochemistry (IHC) and occurs more frequently than in common endometrial cancer, with 25%–30% of cases showing MMR deficiency [5,6,7,8]. MMR-deficient tumors are burdened with somatic mutations due to a defective DNA MMR system. It has been reported that tumors with higher numbers of somatic mutations are more immunogenic and have immune escape mechanisms, such as the programmed cell death-1 (PD-1) and PD-1 ligand 1 (PD-L1) pathways [9,10,11]. Clinical trials of immune checkpoint inhibitors for MMR-deficient tumors have been studied in many carcinomas including colorectal cancer and melanoma [12]. As a biomarker for the effectiveness of immune checkpoint inhibitors, PD-L1 expression on IHC, cytotoxic T lymphocyte (CD8+ T cell), and neo antigen (mutation burden rich) are shown in existing reports [12,13,14,15]. Specifically, when assessing the anti-PD-1 antibody for melanoma, infiltration of CD8+ T cells correlate with response to them [14]. It has been suggested that immune checkpoint inhibitors may be effective when there is a high infiltration of CD8+ T cells into the tumor [16,17,18]. Therefore, immune checkpoint inhibitors are thought to be effective for MMR-deficient tumors. However, the relationship between MMR deficiency and the expression of PD-L1 and CD8+ T cell tumor-infiltration remains poorly understood in DDEC. We hypothesized that prognosis may be improved by the use of immune checkpoint inhibitors in DDEC with MMR deficiency. In the present study, we investigated the relationship between the expression of PD-L1 protein and CD8+ T cell tumor-infiltration and MMR deficiency in DDEC.

## 2. Results

### 2.1. The Clinicopathological Features

The clinicopathological features of the 17 cases of DDEC are summarized in Table 1. The patient ages ranged from 52 to 78 years (median of 62 years). By the International Federation of Gynecology and Obstetrics staging criteria, the number of cases in stages I, II, III, and IV were 5 (29.4%), 1 (5%), 7 (41.2%), and 4 (23.5%), respectively.

### 2.2. Immunohistochemical Findings

The immunohistochemical findings are shown in Figure 1, Figure 2 and Figure 3. Some cases were differentially stained according to the well-differentiated and the undifferentiated components. Out of 17 cases, 11 (64.7%) were MMR-deficient; loss of MMR proteins was observed in the well-differentiated component for 8 cases (MLH1, 8 cases; PMS2, 4 cases; MSH2, 2 cases; and MSH6, none), in the undifferentiated component for 9 cases (MLH1, 6 cases; PMS2, 5 cases; MSH2, 2 cases; and MSH6, 1 case). Overall, 6 cases out of 17 cases had MMR deficiency in both the well-differentiated component and undifferentiated components. Furthermore, 11 cases (64.7%) had PD-L1 expression. PD-L1 expression was observed only in the undifferentiated component (Table 1).

### 2.3. MSI Analysis

We analyzed genomic MSI in 3 cases that were indicated as having MMR deficiency by IHC (Table 1). All cases were considered as MSI-high based on the MSI analysis. In Figure 4, we present case 1 that was evaluated as MSI-high both in the well-differentiated and undifferentiated components.

### 2.4. Statistical Analyses 

Expression of PD-L1 was significantly associated with MMR deficiency in the undifferentiated component (*p* = 0.026; Table 2). In contrast, none of the cases showed PD-L1 expression in the well-differentiated component. In the undifferentiated component, the MMR-deficient group had more CD8 positive T cell infiltration than the MMR-proficient group (*p* = 0.026; Table 3). In the well-differentiated component, there was no significant difference between CD8 positive T cell infiltration and MMR deficiency (*p* = 0.772; Table 4).

## 3. Discussion

DDEC is rare, occurring in only 9% of all endometrial carcinomas [4]. The Cancer Genome Atlas stratifies endometrial carcinomas into four distinct molecular groups on the basis of molecular genetic alterations, namely those with Defective DNA polymerase ε (*POLE*) mutations, those with MSI, those with low copy number alterations, and those with high copy number alterations, including *p53* mutations. However, it is not explicitly specified as to which molecular group DDEC is classified to [6]. The above study suggested that progression free survival is better in patients with POLE mutations than in those with MSI. However, recent studies have reported that MSI is associated with several poor prognostic indicators that are routinely used to make decisions for adjuvant therapy use [19,20,21,22]. Previous studies have suggested that DDEC is associated with a deficiency of MMR proteins (MLH1, PMS2, MLH2, and MSH6) [5]. In recent reports, the prevalence of MMR deficiency in endometrial carcinoma is 25%–30% [6,7,8]. DDEC is generally associated with 58% MMR deficiency and is more frequent than endometrial carcinoma [5]. In addition, DDEC has a poorer prognosis as compared with Grade 3 endometrial carcinoma [1,2,3]. Therefore, individual treatment strategies for DDEC, especially MMR-deficient cases, need to be devised. In recent years, it has been reported that tumors with higher numbers of somatic mutations (high mutation burden), such as MSI tumors, are more immunogenic, and immune checkpoint inhibitors are effective for such tumors [10,23,24]. Although the number of cases was small, Hussaini et al. and Liu et al. demonstrated PD-L1 expression in DDEC [25,26]. Therefore, we hypothesized that prognosis may be improved through the use of immune checkpoint inhibitors (anti PD-1/PD-L1 antibodies) in DDEC with MMR deficiency. Recently, we reported that MMR deficiency is a biomarker for predicting the effect of immune checkpoint inhibitors using immunostaining in endometrial carcinoma [27]. In other reports, tumor infiltration of lymphocytes was associated with the responsiveness of immune checkpoint inhibitors [15,16,17]. From there, we thought that it could be a target for immune checkpoint inhibitors by correlating the tumor infiltration of lymphocytes with MMR deficiency and the expression of PD-L1. Based on this, in the present study, we investigated the expression of PD-L1 and the level of tumor-infiltrating CD8 positive T cells in endometrial carcinoma cases. In this research, MMR deficiency observed in either undifferentiated or well-differentiated components was found to be 64.7%, which is consistent with previous reports [5]. Our results showed that MMR deficiency was significantly associated with PD-1 expression (*p* = 0.026) and the presence of tumor-infiltrating lymphocytes (CD8+) (*p* = 0.026). Our results suggest that DDEC could therefore be a target for immune checkpoint inhibitors.

DDEC is a very rare and new histopathological concept; as such, the molecular mechanisms are poorly understood [1,3,4]. A recent study reported that the evidence regarding similar mutations in the well-differentiated and undifferentiated components suggests that this tumor progresses from a low-grade endometrioid adenocarcinoma to an undifferentiated carcinoma [28]. Interestingly, even within the same tumor, staining results were different in the undifferentiated and well-differentiated components, and PD-L1 was expressed only in the undifferentiated component in our results. Yokomizo et al. first demonstrated the loss of MMR protein expression in the undifferentiated component [29]. These findings may suggest that DDEC has intra-tumor heterogeneity. Wu et al. showed that the metastatic histology of DDEC was mainly composed of an undifferentiated component [1]. Although the reason for DDEC aggressiveness is not clear, it is most likely due to the undifferentiated component. Although this is a hypothesis, an immune escape mechanism occurs during the process of dedifferentiation. As a treatment strategy for DDEC with MMR deficiency, conventional paclitaxel plus carboplatin and cisplatin plus doxorubicin therapies are effective for the well-differentiated component. However, by using immune checkpoint inhibitors in combination with conventional chemotherapy, it may be possible to control the growth of the undifferentiated component and finally lead to the improvement of prognosis. However, this study is limited by the relatively low number of cases due to the rarity of these tumors. In vitro cytotoxicity assays are needed to determine the actual effectiveness of immune checkpoint inhibitors for DDEC. However, in the future, it is hoped that efficacy assessments will be re-evaluated by accumulating clinical trial results based on other carcinoma investigations.

Recent molecular studies have reported that the inactivation of core components of the switch/sucrose non-fermentable (SWI/SNF) chromatin remodeling complex proteins, BRG1 inactivation, INI1 inactivation, or ARID1A/ARID1B co-inactivation are associated with histological dedifferentiation [30,31,32,33]. In these reports, the patients with dedifferentiated or undifferentiated endometrial carcinoma with SWI/SNF complex deficiency were defined as a highly aggressive subset [32]. It was suggested that therapies targeting chromatin remodeling resulting in epigenetic control might be effective [32]. Although undifferentiated carcinoma was included in the study population in this report, only one out of 34 cases had a *POLE* mutation [32]. In other studies, seven out of 13 DDEC cases had *POLE* exonuclease domain mutations [34]. Furthermore, the patients with *POLE*-mutated dedifferentiated and undifferentiated endometrial carcinomas had a favorable outcome. On the other hand, DDEC tended to be associated with MSI due to abnormalities in MLH1/PMS2 [34]. A case of DDEC with MLH1 promoter hypermethylation, high MSI status, and high PD-L1 expression was reported very recently [26]. Our results also showed that most MSI cases were deficient in MLH1/PMS2. In patients with colorectal cancer, tumors with a poor differentiation, Crohn-like lymphoid reaction, and PD-L1 expression occurred more frequently in sporadic MSI cases than in Lynch syndrome-associated cases [35]. In the present study, we did not examine MLH1 promoter methylation status, MMR germline status, *POLE* mutation, or mutation burden status. However, the current results and a recent case report [33] speculated that DDEC had a high mutation burden, resulting in high immunogenicity. Therefore, a pathological diagnosis of DDEC is a potential predictive factor for a good response to immunotherapy targeting. So far, there is little consideration for histopathological-specific therapies. In cases of histopathological specificity according to a special clinical course like DDEC, it is necessary to find individual treatment strategies. To that end, if possible, more cases should be investigated, and the development of cancer genomic medicine using sequencing technology for DDEC in a clinical setting should be further evaluated [36].

In summary, the current results that indicated a high expression of PD-L1 and CD8 positive T cells in the dedifferentiated component of MMR deficient tumors suggest that DDEC deficiency could be a target for immune checkpoint inhibitors (anti PD-1/PD-L1 antibodies), and the presence of MMR deficiency may be a biomarker for a good response to PD-L1 immunotherapy in DDEC.

## 4. Materials and Methods

### 4.1. Study Samples

We searched the pathology databases of Shimane University, Shimane Prefectural Central Hospital, Seirei Hamamatsu General Hospital from 2007 to 2017. Samples were collected from 17 patients who were diagnosed with low-grade endometrioid carcinoma (Grade 1–2) that contained an undifferentiated component. Patients were diagnosed based on the 2014 World Health Organization (WHO) Classification of Tumors of the Female Genital Organs. Some patients were diagnosed with high-grade endometrioid carcinoma [37]. Furthermore, we evaluated the distinction between particularly confusing DDEC and high-grade endometrial carcinoma with reference to the literature by Han et al. Specifically, the undifferentiated component of DDEC represented a solid sheet constructed by tumor cells lacking intercellular adhesion without glandular formation. Furthermore, DDEC has clear boundaries for undifferentiated and well-differentiated components. In contrast, high-grade endometrial carcinoma has a glandular structure and often shifts gradually from low grade [38]. All cases were independently diagnosed by pathologists (Yoshihiro Otsuki and Hideyuki Onuma) at each hospital and reviewed by a gynecologic pathologist (Noriyoshi Ishikawa). Samples were collected after obtaining written consent from all patients with the approval of the Facility Ethical Committee (Shimane University Hospital, Izumo, Japan; approval No. 2004–0381, 5 March 2007).

### 4.2. Immunohistochemistry

The expression of MMR proteins (MLH1, PMS2, MSH2, and MSH6), CD8, PD-L1, were examined by immunostaining. The method used for immunostaining was described in detail in our previous report [27]. Briefly, formalin-fixed and paraffin-embedded sections (4-µm thick) were dewaxed in xylene and hydrated in graded alcohol. After antigen retrieval in a sodium citrate buffer, slides were incubated overnight at 4 °C with antibodies against MLH1 (1:50; Dako, Santa Clara, CA, United States), PMS2 (1:40; Dako, Santa Clara, CA, United States), MSH2 (1:50; Dako, Santa Clara, CA, United States), MSH6 (1:50; Dako, Santa Clara, CA, United States), CD8 (1:100; Roche, Basel, Switzerland), PD-L1 (ab205921, Abcam, Cambridge, CAM, United Kingdom). Immunostaining was evaluated using a double-blind method by two researchers (R.O. and K.N.). We evaluated the differentiated and undifferentiated components of each case. Tumors were considered to be MMR deficient if at least one of the four MMR proteins (MLH1, PMS2, MSH2, or MSH6) was deficient. The level of tumor infiltrating lymphocytes was classified into four categories by CD8 expression: 0, undetectable; 1+, weakly positive (percentage of CD8 positive cells per tumor cells 0–30%); 2+, moderately positive (30−60%); and 3+, strongly positive (≥60%). Cases that were 2+ or 3+ were counted as positive in our analysis. Based on the cut-off value used in many clinical trials including Nivolumab’s 3rd clinical trial on malignant melanoma, IHC of PD-L1 was evaluated as positive if more than 5% of the tumor cells were stained [39,40].

### 4.3. DNA Extraction and MSI Analysis

We performed MSI analysis for three cases that were indicated as MMR deficient according to IHC. DNA was extracted from formalin-fixed paraffin-embedded (FFPE) tissues according to protocols for the isolation of total DNA. Well-differentiated components and undifferentiated components were separately collected macroscopically with reference to Hematoxylin-Eosin (HE) staining and interstitial tissue was collected to use as a control in the analysis. We digested tumor tissues (0.01 M NaCl; 0.5 M Tris-HCl, pH 8.0; 20 mM EDTA; 0.05% Tween-20; 0.1 mg/mLproteinase K) for 1 h at 56 ˚C or until the sample indicated complete lysis. To inactivate the proteinase K, we heated the tissues to 90 ˚C for 1 h. Following this, DNA was extracted with phenol/chloroform treatment and ethanol precipitation. The MSI status was determined using eight microsatellite markers (BAT25, BAT26, D2S123, D5S346, D17S250, NR21, MONO27, and NR2). We analyzed the amplicons on the ABI PRISM 310 Genetic Analyzer and evaluated allelic sizes by GeneMapper (Appleid Biosystems, Thermo Fisher K.K Yokohama Japan). Tumors with instability at two or more markers were considered as MSI-high.

### 4.4. Statistical Analyses

The relationships between MMR status and the expression of CD8 and PD-L1 were assessed using a Chi-squared test. We examined the well-differentiated and undifferentiated components separately.

## Figures and Tables

**Figure 1 ijms-20-03744-f001:**
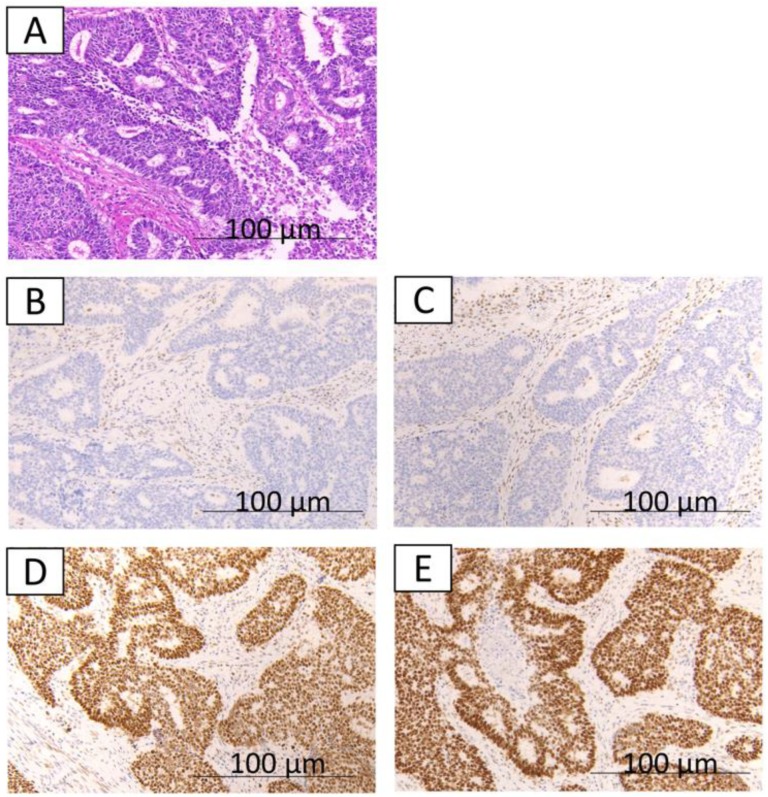
The immunohistochemical findings from case 1. **A,B,C,D,E**: well-differentiated component; hematoxylin and eosin staining show well-differentiated endometrioid glands (**A**); loss of expression of MLH1 (**B**); loss of expression of PMS2 (**C**); expression of MSH2 (**D**); expression of MSH6 (**E**).

**Figure 2 ijms-20-03744-f002:**
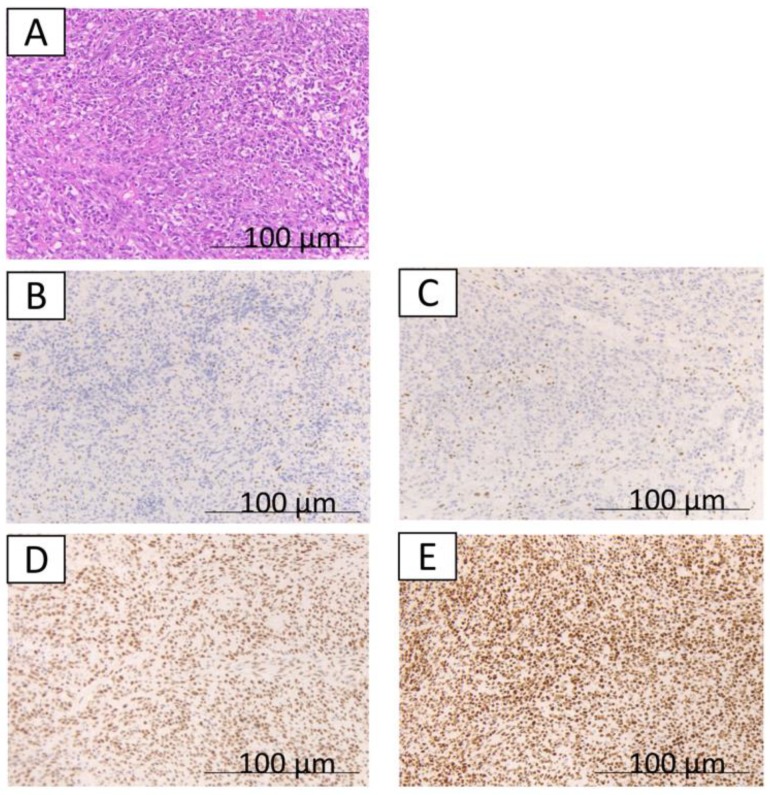
The immunohistochemical findings from case 1. **A,B,C,D,E** indicate a typical undifferentiated component; hematoxylin and eosin staining show the undifferentiated component (**A**); loss of expression of MLH1 (**B**); loss of expression of PMS2 (**C**); expression of MSH2 (**D**); expression of MSH6 (**E**).

**Figure 3 ijms-20-03744-f003:**
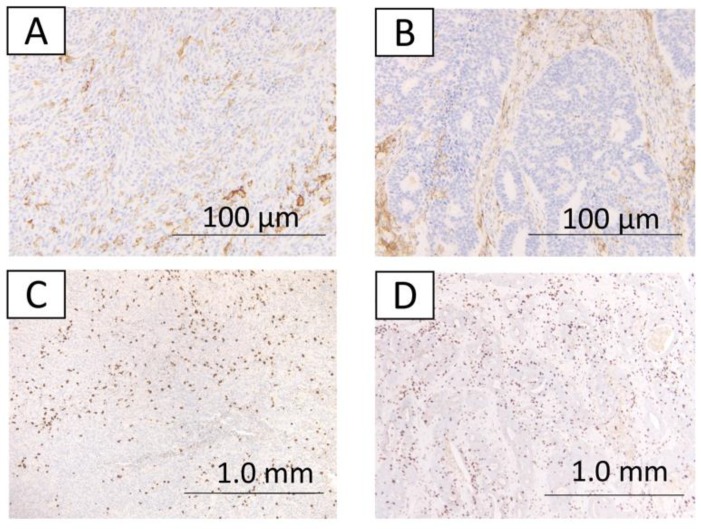
The immunohistochemical findings from case 1. **A,B,C,D** indicate no expression of PD-L1 in the well-differentiated component (**A**). Expression of PD-L1 in the undifferentiated component (**B**). CD8 expression score of 2 in the well-differentiated component (**C**). CD8 expression score of 2 in the undifferentiated component (**D**).

**Figure 4 ijms-20-03744-f004:**
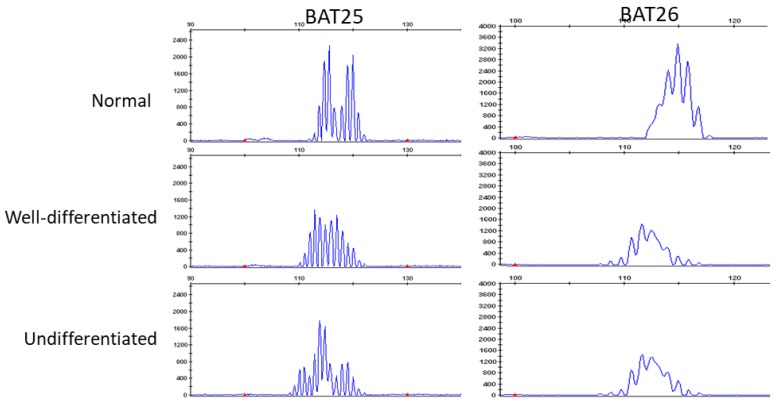
Microsatellite instability (MSI) analysis of normal (**top**), well-differentiated components (**middle**), and undifferentiated components (**bottom**). Two microsatellite markers (BAT25 and BAT26) show instability and are visible as the shift in well-differentiated and undifferentiated components the size (base pairs) of the amplification products.

**Table 1 ijms-20-03744-t001:** Clinicopathologic features of 17 dedifferentiated endmetorial carcinoma.

Case	Age	FIGO Stage	MLH1		PMS2		MSH2		MSH6		MSI Analysis	PD-L1		CD8	
			WD	UD	WD	UD	WD	UD	WD	UD		WD	UD	WD	UD
1	54	IIIC1	d	d	d	d					MSI-high	negative	positive	1+	1+
2	57	IIIC1	d	d	d	d						negative	positive	2+	2+
3	64	IB	d	d	d	d					MSI-high	negative	positive	3+	3+
4	78	IVB	d	d							MSI-high	negative	positive	2+	2+
5	56	IVB	d	d			d	d				negative	positive	1+	0
6	74	IA	d		d	d						negative	positive	2+	1+
7	58	IB						d				negative	positive	3+	2+
8	55	IIIC2								d		negative	positive	3+	3+
9	63	IIIA										negative	positive	2+	2+
10	57	IB										negative	positive	2+	2+
11	73	IA										negative	positive	2+	1+
12	77	IIIA	d									negative	negative	3+	2+
13	62	IIIC	d				d					negative	negative	1+	1+
14	53	IVB										negative	negative	2+	1+
15	59	IIIA										negative	negative	0	0
16	56	IVB		d		d						negative	negative	2+	0
17	79	II										negative	negative	1+	0

WD. well-differentiated component; UD. undifferentiated component; d. deficient; MSI. microsatellite instability.

**Table 2 ijms-20-03744-t002:** Relationship between status of MMR and PD-L1 expression. (undifferentiated component).

Parameter	MMRd	MMRp	*p*-Value
	*N* = 9	*N* = 8	
PD-L1-no. (％)			0.026
positive	8(88.9)	3(37.5)	
negative	1(11.1)	5(62.5)	

MMRd. Mismatch repair deficient; MMRp. Mismatch repair proficient.

**Table 3 ijms-20-03744-t003:** Relationship between status of MMR and CD8 expression. (undeffirentiated component).

Parameter	MMRd	MMRp	*p*-Value
	*N* = 9	*N* = 8	
CD8-no. (％)			0.026
positive	8(88.9)	3(37.5)	
negative	1(11.1)	5 (62.5)	

MMRd. Mismatch repair deficient; MMRp. Mismatch repair proficient.

**Table 4 ijms-20-03744-t004:** Relationship between status of MMR and CD8 expression. (well differentiated component).

Parameter	MMRd	MMRp	*p*-Value
	*N* = 8	*N* = 9	
CD8-no. (％)			0.772
positive	3(37.5)	4(44.4)	
negative	5(62.5)	5 (55.5)	

MMRd. Mismatch repair deficient; MMRp. Mismatch repair proficient.

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
