# Peer review of "Dedifferentiated Endometrial Carcinoma Could be A Target for Immune Checkpoint Inhibitors (Anti PD-1/PD-L1 Antibodies)"

_ijms, 2019, doi:10.3390/ijms20153744_

Round 1
Reviewer 1 Report
Dedifferentiated endometrioid adenocarcinoma is a rare uterine neoplasm containing both low-grade endometrioid adenocarcinoma and undifferentiated carcinoma. Dedifferentiated endometrioid adenocarcinoma is typically associated with low response rate to the therapy and a poor prognosis. Therefore, new therapies is urgently necessary. R. Yokomizo et al firstly demonstrated the loss of MMR protein expression in the undifferentiated components of these patients, whereas more recently Hussaini al et al similarly to the work under review, demonstrated the co-presence of deficit MMR and PD-L1.Please includes these papers in our manuscript. Please include also in the introduction, another papers describing the use of immune check points in tumor with MMR, for exemple could include the article below,
R. Yokomizo, K. Yamada, Y. Iida et al., “Dedifferentiated endometrial carcinoma: A report of three cases and review of the literature,” Molecular and Clinical Oncology, vol. 7, no. 6, pp. 1008–1012, 2017.
Al-Hussaini M, Lataifeh I, Jaradat I, Abdeen G, Otay L, Badran O, Abu Sheikha A, Dayyat A, El Khaldi M, Ashi Al-Loh S.
Undifferentiated Endometrial Carcinoma, an Immunohistochemical Study
Including PD-L1Testing of a Series of Cases From a Single Cancer Center. Int J Gynecol Pathol. 37(6):564-574; 2018
Immunotherapy for colorectal cancer: where are we heading? Basile, D., Garattini, S.K., Bonotto, M., (...), Cardarelli, N., Aprile, G. Expert Opinion on Biological Therapy 17(6), pp. 709-721; 2017
Author Response
Please see the attachment for reviewers comment 1

Reviewer 2 Report
There are some typos and missing spaces.
The paper detailed dedifferentiated endometrial carcinoma as a defined histology but literature is scarce and the widest series (Silva et al) is composed by 15 cases. Moreover difference between dedifferentiated and mixed EC should be described.There is no extensive description on how DDEC histology has been defined, in particular on how undifferentiated component (UC) has been identified and distinguished from High grade EC (for example if IHC with EMA, CK18, PAX-8 have been performed). Moreover a check from 2 pathologist could be useful in these rare cases. Expression of PDL-1 could be evalutated both in tumor cells and immune ones (Howitt Jama 2015) and expression of PD-1 could be defined.
The title, abstract and discussion could be misleading because expression of PDL-1 or CD8 cannot properly predict response to Checkpoint inhibitors. Moreover percentage of MMR deficiency is not far from high grade ECs in TCGA and it is not typical of DDEC histology. Moreover no in vitro cytotoxicity assay has been performed so it could not be written that " Our results show that immune checkpoint inhibitors (anti PD-L1/PD-1 antibodies) could be effective in the treatment of DDEC, especially in the undifferentiated component" line 31-32.
The work shoud be reviewed extensively focusing on discussion and methods
Author Response
Please see the attachment file for Reviewer comments 2

Round 2
Reviewer 2 Report
the authors addressed the issues. the article is now acceptable